# Specialized Pediatric Palliative Care Services in Pediatric Hematopoietic Stem Cell Transplant Centers

**DOI:** 10.3390/children8080615

**Published:** 2021-07-21

**Authors:** Hilda Mekelenkamp, Teija Schröder, Eugenia Trigoso, Daphna Hutt, Jacques-Emmanuel Galimard, Anne Kozijn, Arnaud Dalissier, Marjola Gjergji, Sarah Liptrott, Michelle Kenyon, John Murray, Selim Corbacioglu, Peter Bader

**Affiliations:** 1Willem-Alexander Children’s Hospital, Department of Pediatrics, Leiden University Medical Center, P.O. Box 9600, 2300 RC Leiden, The Netherlands; 2New Children’s Hospital, P.O. Box 347, 00029 Helsinki, Finland; teija.schroder@gmail.com; 3Paediatric Transplant Unit, Hospital University and Polytechnic Hospital LA FE, Avinguda de Fernando Abril Martorell, 106, 46026 Valencia, Spain; eugeniatrigoso57@gmail.com; 4Division of Pediatric Hematology and Oncology, The Edmond and Lily Safra Children’s Hospital, Sheba Medical Center, Ramat Gan 52621, Israel; daphna.hutt@sheba.health.gov.il; 5EBMT Statistical Unit, CEDEX 12, 75571 Paris, France; jacques-emmanuel.galimard@upmc.fr; 6EBMT Leiden Study Unit, Rijnsburgerweg 10, 2333 AA Leiden, The Netherlands; EBMT_NursesGroup@lumc.nl; 7EBMT Paris Study Unit, CEDEX 12, 75571 Paris, France; arnaud.dalissier@upmc.fr; 8Ospedale Pediatrico Bambino Gesù, Department of Onco-Hematology and Cell and Gene Therapy, Piazza di Sant’Onofrio, 4, 00165 Rome, Italy; marjola.gjergji@opbg.net; 9IEO, European Institute of Oncology IRCCS, Via Ripamonti 435, 20141 Milan, Italy; sarah.liptrott@ieo.it; 10Department of Haematological Medicine, King’s College Hospital NHS Foundation Trust, Denmark Hill, London SE5 9RS, UK; michelle.kenyon@nhs.net; 11Haematology and Transplant Unit, Christie Hospital NHS Foundation Trust, Wilmslow Road, Manchester M20 4BX, UK; j.murray10@nhs.net; 12Department of Pediatric Hematology, Oncology and Stem Cell Transplantation, University of Regensburg, Franz-Josef-Strauss-Allee 11, 93052 Regensburg, Germany; selim.corbacioglu@mac.com; 13Division for Stem Cell Transplantation, Immunology and Intensive Care Medicine, Department for Children and Adolescents, University Hospital Frankfurt, Goethe University, Theodor-Stern-Kai 7, 60590 Frankfurt, Germany; Peter.Bader@kgu.de

**Keywords:** hematopoietic stem cell transplantation, palliative care, palliative care services, pediatric

## Abstract

Hematopoietic stem cell transplantation (HSCT) is widely used in pediatric patients as a successful curative therapy for life-threatening conditions. The treatment is intensive, with risks of serious complications and lethal outcomes. This study aimed to provide insight into current data on the place and cause of death of transplanted children, the available specialized pediatric palliative care services (SPPCS), and what services HSCT professionals feel the SPPCS team should provide. First, a retrospective database analysis on the place and cause of death of transplanted pediatric HSCT patients was performed. Second, a survey was performed addressing the availability of and views on SPPCS among HSCT professionals. Database analysis included 233 patients of whom the majority died in-hospital: 38% in the pediatric intensive care unit, 20% in HSCT units, 17% in other hospitals, and 14% at home or in a hospice (11% unknown). For the survey, 98 HSCT professionals from 54 centers participated. Nearly all professionals indicated that HSCT patients should have access to SPPCS, especially for pain management, but less than half routinely referred to this service at an early stage. We, therefore, advise HSCT teams to integrate advance care planning for pediatric HSCT patients actively, ideally from diagnosis, to ensure timely SPPCS involvement and maximize end-of-life preparation.

## 1. Introduction

Hematopoietic stem cell transplantation (HSCT) is widely used in pediatric patients with malignant and non-malignant disorders [1,2]. HSCT is a successful therapy offering the possibility of a cure for life-threatening conditions; however, it is an intensive treatment with risk of serious complications, significant physical and psychological symptoms, poor quality of life (QoL), and can be lethal [3,4,5,6]. As reported in previous studies, pediatric patients who died following HSCT were often treated in the pediatric intensive care unit (PICU) with invasive treatments during End-of-Life (EOL) [4,6,7,8]. Of these studies, three showed that pediatric HSCT patients often died from treatment-related complications (60, 68, and 81%) compared to disease-related (40, 32, and 19%) [4,6,8]. The place of death of a child with an incurable illness depends on several factors [9] and varies in several studies [10,11]. In the USA between 1989–2003, the place of death for children whose deaths were attributed to complex chronic conditions increasingly shifted from predominantly hospital to home setting [10]. In contrast, a large cross-national study among 11 countries reported that the hospital was the most common place of death for children with complex chronic conditions [11]. It must be noted here that the latter study showed considerable country variation in home deaths, particularly for children with malignancies. Preferred place of death was recently endorsed as an important quality measure for EOL care in pediatric patients [12]. Achievement of goal-concordant EOL care, specifically the location of death, is important [13]. Palliative care involvement has been shown to improve concordance rates between patient/family goals and place of death [14]. 

Early identification, as well as the assessment and treatment of symptoms (whether physical, psychosocial, or spiritual in origin), are vital approaches for palliative care to prevent and relieve suffering [15]. Integration of palliative care during hospitalization for HSCT has proved beneficial in adult care, improving patients’ QoL and symptom burden during HSCT [5,16]. Pediatric palliative care, as defined by the WHO, not only entails the active total care of the child’s body, mind, and spirit, and involves giving support to the family [15]. Children who died after HSCT were reported to suffer more from physical and psychological symptoms compared with children who had not undergone HSCT [8]. The authors of this study noted that the child’s condition can deteriorate fast while the treatment is still focused on survival, resulting in less time to prepare for EOL [8]. Parent reports showed that, in hindsight, parents experienced concerns regarding their child’s suffering [17] and were at greater risk of decreased psychological wellbeing [18,19]. In contrast, studies have shown that transplanted children, adolescents, and young adults who received (pediatric) palliative care were more likely to die outside the (P)ICU; this reflects an impact of palliative care involvement on a patient’s location of death [4,6]. Additionally, those who received palliative care were less likely to have intervention-focused care, and more likely to have opportunities for EOL communication and advance preparation [4,6]. For parents too, it is becoming increasingly clear that palliative care support is instrumental in preparing for their child’s EOL, reducing levels of parental distress and long-term parental grief [20,21]. Taken together, palliative care can play a beneficial role for all parties involved in terms of course of treatment, maximizing QoL, and coping with loss.

The high-risk nature of HSCT means that there is a potential for transplant teams to be regularly confronted with having to provide EOL care. The intensity of treatment and the anticipation of cure may impede the provision and initiation of satisfactory palliative care. HSCT teams may be reluctant to discuss a patient’s risk of dying or EOL decisions before a child’s clinical deterioration because they feel this may infer a lack of commitment by the HSCT team or diminish the child’s/families hope for a positive outcome [22]. Specialized pediatric palliative care services (SPPCS) could support an early palliative care approach. In Europe, access to SPPCS varies considerably per country. The ideas about the provision and the organization of SPCCS in clinical HSCT practice are presently unknown. It is necessary to determine what services are deemed important by transplant centers and what services are available to children and the family undergoing HSCT. We aim (1) to describe the place and cause of death of children who died after receiving an HSCT, (2) to identify available SPCCS in clinical HSCT practice and explore what service health care professionals (HCPs) feel the SPCCS team should provide.

## 2. Materials and Methods

This descriptive study was conducted in two consecutive parts. The first part consisted of a retrospective database analysis on the place and cause of death of pediatric patients after HSCT. The second part was a survey investigating views of health care professionals (HCPs) on specialized pediatric palliative care services (SPPCS).

### 2.1. Database 

To answer the first aim, data was accessed regarding the place and cause of death of transplanted pediatric patients who died between 2009 and 2015. These data were derived from the patient registry that the European Society for Blood and Marrow Transplantation (EBMT) maintains and to which member centers register their transplanted patients. Member centers were asked to confirm the details and to complete incomplete data. Data on age, sex, disease, age at death, place, and cause of death were requested. 

Responses to the cause of death and the underlying disease were categorized before analysis (see legend Table 1). Quantitative variables are described as median, first and third quartile, minimum and maximum. Qualitative variables are described as numbers and percentages. Chi-square or Fisher tests were used to test the association between qualitative variables and the place of death. Kruskal–Wallis test was used to test the association between quantitative variables and place of death. The significance level was fixed at 0.05 and all *p*-values are two-sided. All analyses were done using the open-source statistical software R, version 4.0.4. [23].

### 2.2. Survey

To answer the second aim, a survey was conducted among HSCT professionals working in EBMT member centers to evaluate what SPPCS are currently available in clinical practice and what services HCPs feel should be provided.

The survey was developed by an international team including a pediatric HSCT physician and pediatric HSCT nurses, supported by a data manager and statistician. The survey consisted of closed-ended questions to HSCT professionals regarding (1) what palliative services are currently available for pediatric patients undergoing HSCT, (2) what aspects do HCPs consider important for palliative care services to offer pediatric patients undergoing HSCT, and (3) a rating of the importance of services offered by SPPCS on a 5-point Likert scale (extremely important (+2); very important (+1); neutral (0); not very important (−1); not necessary (−2)). This survey was first distributed in 2015 via an online survey platform (Survey Monkey^®^, San Mateo, CA, USA). Reminders were sent in 2015, 2016, and 2018. We asked one physician per center to complete the first part of the questionnaire and additionally a transplant nurse and another HCP working in the HSCT unit (e.g., social worker, play therapist, psychologist, spiritual advisor) to complete the second part of the questionnaire.

Survey answers are described as numbers and percentages of the whole group and by subgroups (physicians, nurses, and other HCPs). 

## 3. Results

### 3.1. Database 

For the retrospective database analysis, eleven EBMT member centers from eight countries (Belgium, France, Israel, The Netherlands, Spain, Sweden, Switzerland, and the United Kingdom) participated. Two hundred thirty-three pediatric patients were included, of which 147 (63%) had a malignant disease and 86 (37%) a non-malignant disease (Table 1 and Table 2). The median age of death was 7.2 years (IQR 3.3–13.2) and the majority were male (60%). Of all patients, 76% underwent an HSCT once, 19% twice, 4% three times, and 1% four times. The last transplantations took place between 2009 and 2014. Patients died with a median of 3.8 months (IQR 1.6–9.1) after their last HSCT. Patients who had a non-malignant disease were younger at their first HSCT (*p* < 0.0001), died earlier after their last HSCT (*p* < 0.0001), and were younger when they died (*p* < 0.0001) (Table 2).

#### Place of Death and Cause of Death

Most of the patients died in-hospital (75%), followed by 38% in PICU, 20% in the HSCT unit, and 17% in other hospitals. A small percentage of the patients died (14%) at home or in a hospice and the place of death was reported as unknown by centers for 11% of the patients. The place of death in relation to the time between last HSCT and death, cause of death, and the underlying reason for HSCT (malignant/non-malignant) were significantly different (*p* < 0.0001) (Table 1). Patients who died at home or in a hospice, died later after HSCT with a median of 8.7 months (IQR 4.1–17.4) after their last HSCT, compared to patients who died in-hospital (PICU: 1.8 months (IQR 0.8–4.4); HSCT-unit: 2.6 months (IQR 1.4–4.4); other hospitals: 4.8 months (IQR 3.6–11.4); (for an unknown place of death: 8.2 months (IQR 3.8–17.7)), see Figure 1. 

Of all the patients who died in hospital, the majority died of HSCT-related complications compared to disease-related complications. Of the patients who died from HSCT-related complications, over 90% died in-hospital (PICU, HSCT unit, or other hospitals) (Table 1). Place of death ratios were different for patients who died from disease-related complications, of whom about 50% died in-hospital (PICU, HSCT unit, or other hospitals) (Table 1). Most of the patients who died outside of the hospital had a malignant disease as an underlying disease compared to non-malignant diseases. Of the patients who had a malignancy, more patients died in the PICU compared to the HSCT unit or other hospitals (Table 1). The cause of death for patients who had a non-malignant disease was more often HSCT-related compared to disease-related (*p* < 0.0001) (Table 2).

### 3.2. Survey 

A total of 98 HCPs from 54 centers in 23 countries participated in the survey. Of these respondents, the majority were residents in European countries (91%), compared with non-European countries (9%) (Appendix A). Of the participating centers, 63% were JACIE [24]-accredited. The total group included physicians (43%), nurses (41%), and affiliated HCPs (16%). The majority of the total group of HCPs was female (80%), and most of the HCPs were > 40 years old (70.5%). Of the HCPs, 30% had more than 20 years of work experience, followed by 22% with less than five years of experience. The other groups with 5–10, 10–15, and 15–20 years of experience showed comparable percentages of 16, 17, and 16%, respectively. (Table 3).

Nearly all HCPs (92%) indicated that HSCT patients, in light of the risk of transplant-related mortality and relapse, should have access to SPPCS (Appendix A) and 83% of the physicians answered that their team had access to such a service. Only 44% of the physicians routinely referred children to an SPPCS (Appendix A).

When HSCT teams routinely refer to an SPPCS (*n* = 18), the team composition included at least a physician dedicated to palliative care (94%), a pain specialist (94%), a nurse dedicated to palliative care (87%), a play therapist (87%), a social worker (81%), an educationalist (64%), a pastor or other spiritual specialist (60%) and an ethicist (23%). When HSCT-teams did not routinely refer to SPPCS (*n* = 23), palliative care was provided by the individual transplant physician in charge (95%), by an adult palliative care service (22%), by a nurse-led team (39%), or by a variety of others specialties such as pain teams or psychologists (50%) (Appendix A).

Of the physician respondents, 71% answered that their HSCT unit had access to an out-of-hours palliative care team (referring to a palliative care team available outside office hours), 59% of the HSCT units had access to a pediatric hospice, and 56% to specialized bereavement services (Appendix A). Of the HCPs, 24% considered an initial referral to an SPCCS appropriate at the onset of terminal care, 21% considered referral at the time of difficult symptom control, and 20% at the time of original diagnosis. Few HCPs (13%) answered that this referral to SPPCS should be at the onset of a life-threatening event, at the time of admission to the HSCT unit (13%), or at the time of admission to PICU (4%). Almost nobody thought that referral was unnecessary (3%) or that referral should be initiated at the time point of providing bereavement counseling (1%) (Appendix A). Only 40% of the physician respondents routinely used advance care planning (ACP).

Possible services provided by an SPPCS were ranked for importance (Appendix A and Figure 2). All services were ranked mainly as very important and important. Pain management, other symptom control, emotional support, facilitating communication with parents about EOL issues, and facilitating shared decision making were ranked as the most important services of the SPPCS. All these services were indicated as (very) important by over 90% of the respondents. Ethical guidance and spiritual support were ranked as the less important services, with 60–80% of the respondents indicating these services as (very) important. The different groups of HCPs rated on the possible SPPCS similarly, with higher rankings generally observed among nurses and other HCPs compared to physicians.

## 4. Discussion

This study provides important insight into pediatric palliative care in the HSCT setting. The data revealed that most pediatric patients who died following HSCT, did so in-hospital, mainly in PICU and HSCT units. The subgroups of pediatric patients who died shortly after their HSCT, compared to those who died later on following beyond HSCT, died more often in the PICU. Likewise, patients who had a non-malignant disease, compared to those who had a malignant disease, more often died in the PICU. The same applies to the group of patients who died from HSCT-related complications rather than disease-related complications. With regard to the views of the HCPs on SPCCS, an important observation in this study is that most of the participating HCPs expressed that patients should have access to SPPCS. However, most respondents indicated having access to SPPCS, not all teams routinely referred to this service. HCPs differed in opinion about the best moment for referral to SPPCS. Nevertheless, the consultation of the SPPCS team is important for HCPs for many issues and most of all for pain management.

Our study showed that pediatric patients who died shortly after an HSCT often died in-hospital and frequently in the PICU, especially those who died from treatment-related complications. The time frame and place of death seem related to each other and supported the findings of a previous study [4]. Two previous studies showed that transplanted patients compared to non-transplanted patients who had malignant diseases more often died in PICU [4,8]. Our study additionally shows that HSCT patients transplanted for non-malignant diseases compared to those who were transplanted for malignant diseases died more often in the PICU. In contrast, patients with malignancies more often died at home. This observed distinction in place of death validates another report in a broader population, where patients with malignancies were more likely to die at home compared to patients with other chronic conditions; however, these practices differed between countries [11]. In our opinion, this difference in place of death could be explained by the fact that the treatment trajectories of patients with malignant diseases are more commonly known as palliative after being confronted with disease-related complications such as relapse or progression. Adopting such a palliative care approach can make it possible to prepare for EOL and choose a location for continuing care outside of the hospital. Additionally, we have experienced that the treatment trajectories of the heterogeneous group of non-malignant patients are different and less predictable, also for the EOL phase. Our results showed that patients who had a non-malignant disease more often died from HSCT-related complications. These complications often need highly specialized cure-focused care, including PICU, [6,8] suggesting less predictable pathways. Qualitative research involving bereaved parents identified several factors that affect parents when deciding on the place of EOL care, among others, awareness of the child’s dying and timing of the decision-making [9]. Since the EOL care-pathway is difficult to predict, especially in the case of HSCT-related complications, the timing and opportunity of EOL decision-making may come too late or may easily be missed.

We demonstrated that among 98 HSCT-HCPs, 83% had access to an SPCCS. The availability is quite similar to a previous study surveying for SPCCS among 142 HCPs and showing 75% availability [25]. Our study showed that only 13% of the HCPs considered an initial referral to an SPPCS appropriate at the time of admission for HSCT. In the survey reported by Weaver et.al., 31% of the respondents reported HSCT as a trigger for referral [25]. This finding of lower referral rates may be explained by the fact that the HSCT physician in charge often provided palliative care. In our opinion, both the extensive experience HSCT teams have with palliative care and the often long-term relationship with the patient can contribute to less urgency in seeking consultation from the SPCCS team by the HSCT team. A previous survey among HSCT physicians has provided the insight that a certain amount of the HCPs feel that the palliative care team did not have enough understanding of the specific target group [26]. Additionally, HSCT physicians expressed concerns about how patients perceive palliative care and may lose hope for a cure as a result [26]. The literature describes several barriers for accessing palliative care in pediatric oncology at the level of policy, the health system, organizations, and individual providers [27]. For individual providers, these barriers include a lack of knowledge, discomfort with speaking about death, cultural differences, and a lack of time or an established process to integrate palliative care services like ACP [27,28]. We found that HCPs valued all surveyed services of an SPCCS-team as highly important, underlining adopting the concept of palliative care as a holistic approach as stated in the WHO definition.

This study has some strengths and limitations. The strength is the ability to analyze patient data on place of death from over 200 patients from eight countries. Furthermore, the survey was developed and filled in by a group of international HCPs, improving the generalizability. Unfortunately, due to a certain amount of ‘unknown places of death’ (which cannot be assigned) and one of the subgroups being too small, it was not possible to perform a multivariate analysis to assess the impact of predictor variables on the place of death. Possible data bias cannot be ruled out, since survey answers showed a trend towards a positive palliative care approach. Our dataset contains only one outlier, who answered most aspects on palliative care as not important. It is questionable whether all HCPs are positive about SPPCS in general or if respondents to our survey were already interested in and shared positive opinions about palliative care. Lastly, additional research is needed to provide insight into the observed discrepancy between the perceived value of SPCCS and the referral rates. We are also aware that interest in palliative care, as a single or dual approach together with curative care, has grown in recent years and may impact findings. Still, our data solidifies the importance of integrating palliative care into current practice for HSCT professionals.

Our findings encourage several clinical implications. Children who were transplanted for a non-malignant disease and had HSCT-related complications died mainly in the PICU, an acute care setting, and it appears there was less preparation for the EOL and death care. To increase preparation time, we believe awareness needs to be raised within HSCT teams of a two-track pathway for their patients. Early integration of palliative care, defined as care focused on QoL and reducing suffering, should be integrated with standard HSCT care. In such a concurrent care pathway, the needs and wishes of the patient and family can be integrated ideally from the main diagnosis or at least from the start of the HSCT (often referred to as ACP). For HSCT teams, this could be challenging because their care is very much focused on cure and it is relatively common in their practice that difficult times for the patient and family are part of the trajectory. Therefore, we suggest an active approach for HSCT teams. Firstly, ACP skills should become routine [29,30] and ACP needs to be implemented in and supported by the organization [27,28]. Secondly, HSCT teams should consult the SPPCS from an earlier stage in the trajectory to work together to approach care from a multidisciplinary perspective [5,16,28,31]. Early palliative care integration in HSCT is previously described in the literature, suggesting a custom-made, integrative model combining intensive curative care with palliative care focused both on cure and increasing QoL [31]. Strategies to reach such an early integration could vary from SPPCS team consultation initiated by the HSCT team, consultation based on triggers, universal screening consultations, standard palliative care consultations for all HSCT patients, integrating the SPPCS into the multidisciplinary team, or incorporation of ACP in the HSCT trajectory [5,27,31,32,33,34]. In such a cooperative setting the SPPCS team can reinforce the care given by the HSCT teams.

## 5. Conclusions

In conclusion, pediatric HSCT patients often die in the PICU, a setting where there is less time to prepare for the EOL. Based on our results, we recommend integrating ACP in the HSCT trajectory and clinical partnership with the SPCCS in the care for pediatric HSCT patients. More research is needed on the integration of ACP at an early stage in the HSCT trajectory. Furthermore, the observed discrepancy between the perceived value of SPCCS and the referral rate by HCPs warrants further investigation into the collaboration between HSCT and SPCCS teams.

## Figures and Tables

**Figure 1 children-08-00615-f001:**
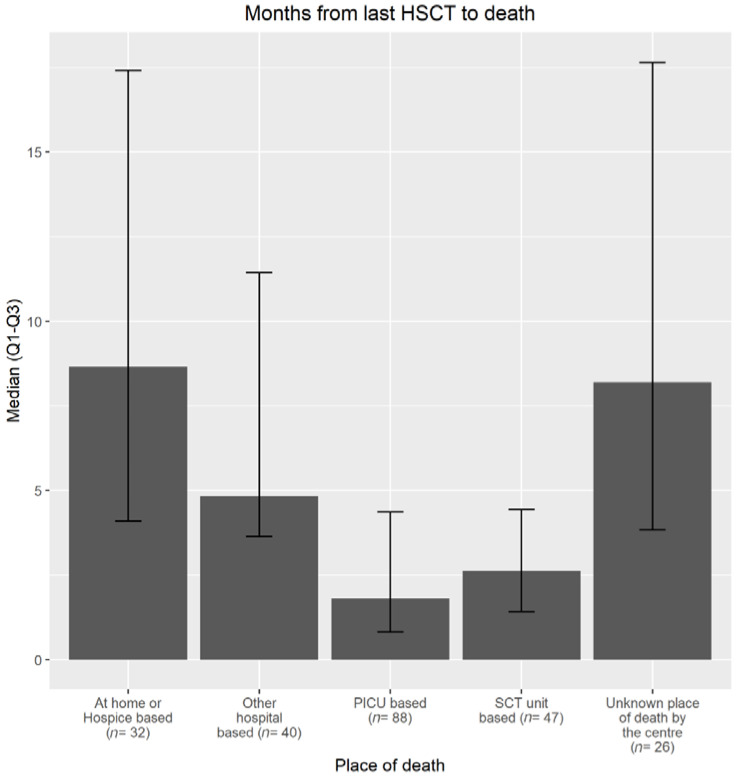
Time between last HSCT and death.

**Figure 2 children-08-00615-f002:**
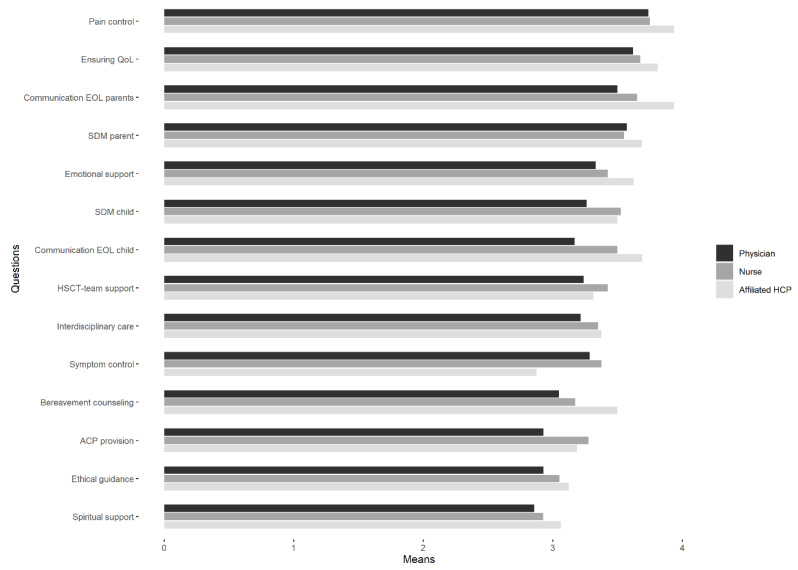
Services HSCT professionals feel the SPPCS team should provide, ranked by importance (i.e., group mean estimation for each service). Abbreviations: ACP = Advance Care Planning; HSCT = Hematopoietic Stem Cell Transplantation; EOL = End-of-Life; SDM = Shared Decision-making; QoL = Quality Of life; HCP = Health Care Professional.

**Table 1 children-08-00615-t001:** Patients’ characteristics grouped by place of death.

Variables	Modalities	All Patients (*n* = 233)	At Home or Hospice (*n* = 32)	PICU (*n* = 88)	Other Hospital (*n* = 40)	HSCT Unit (*n* = 47)	Unknown Place of Death by the Center (*n* = 26)	Test *p*-Value
Age at first HSCT	median [IQR]	6.01 [2.6–11.8]	7.9 [3.1–10.7]	5.7 [1.7–12.8]	4.8 [2.1–11.8]	6.9 [2.3–10.8]	7 [3.5–11.6]	0.6
	(range)	(0.2–18)	(1.1–17.8)	(0.2–18)	(0.2–17.4)	(0.4–15.9)	(0.6–17.9)	
Patient sex *n* (%)	Female	94 (40.3)	15 (16.0)	37 (39.4)	10 (10.6)	18 (19.1)	14 (14.9)	0.16
	Male	139 (59.7)	17 (12.2)	51 (36.7)	30 (21.6)	29 (20.9)	12 (8.6)	
Time between last HSCT and death (in month)	median [IQR]	3.8 [1.6–9.1]	8.7 [4.1–17.4]	1.8 [0.8–4.4]	4.8 [3.6–11.4]	2.6 [1.4–4.4]	8.2 [3.8–17.7]	<0.0001
	(range)	(0–48.9)	(1.6–48.9)	(0.1–37.1)	(0.6–47.5)	(0–28.8)	(1.8–36.8)	
Age at death (in years)	median [IQR]	7.2 [3.3–13.2]	8.9 [4.9–13.1]	6.9 [2.6–13.2]	6.5 [3.6–12.6]	7.4 [2.5–12.3]	8.7 [4.2–12.7]	0.48
	(range)	(0.3–21.6)	(1.8–20.2)	(0.3–21.6)	(0.6–19)	(0.6–17.9)	(2.5–19.5)	
Cause of death * *n* (%)	HSCT-related	137 (59.3)	5 (3.6)	76 (55.5)	15 (10.9)	34 (24.8)	7 (5.1)	<0.0001
	Disease-related	94 (40.7)	27 (28.7)	12 (12.8)	24 (25.5)	13 (13.8)	18 (19.1)	
	missing	2	0	0	1	0	1	
Total number of HSCT *n* (%)	Single HSCT	176 (75.5)	28 (15.9)	68 (38.6)	31 (17.6)	29 (16.5)	20 (11.4)	0.11
	Multiple HSCT	57 (24.5)	4 (7.0)	20 (35.1)	9 (15.8)	18 (31.6)	6 (10.5)	
Disease ** *n* (%)	Malignant	147 (63.1)	29 (19.7)	47 (32.0)	26 (17.7)	21 (14.3)	24 (16.3)	<0.0001
	Non-Malignant	86 (36.9)	3 (3.5)	41 (47.7)	14 (16.3)	26 (30.2)	2 (2.3)	

Abbreviations: HSCT = Hematopoietic Stem Cell Transplantation; PICU = Pediatric Intensive Care Unit * Responses were categorized into HSCT-related and disease-related. Original responses included >80 different responses related to original disease or HSCT complications. ** Responses were categorized into Malignant and Non-Malignant disease. Original responses included: Acute Leukemia, auto-immune disease, bone marrow failure, chronic leukemia, hemoglobinopathies, histiocytic disorders, inherited disorders, lymphoma, myelodysplastic/myeloproliferative, and solid tumors. Answers are described as number (and percentage by row conditionally for the patient sex, cause of death, total number of HSCT and disease) or median [IQR].

**Table 2 children-08-00615-t002:** Patients’ characteristics grouped by disease (malignant/non-malignant).

Variables	Modalities	All Patients (*n* = 233)	Malignant (*n* = 147)	Non-Malignant (*n* = 86)	Test *p*-Value
Age at first HSCT	median [IQR]	6.0 [2.6–11.8]	7.5 [4–13]	2.7 [0.7–8.4]	<0.0001
	(range)	(0.2–18)	(0.5–17.9)	(0.2–18)	
Patient sex *n* (%)	Female	94 (40.3)	59 (40.1)	35 (40.7)	0.93
	Male	139 (59.7)	88 (59.9)	51 (59.3)	
Time between last HSCT and death (month)	median [IQR]	3.8 [1.6–9.1]	4.8 [2.7–10.9]	2.1 [1.1–4.2]	<0.0001
	(range)	(0–48.9)	(0.1–48.9)	(0–47.5)	
Age at death	median [IQR]	7.2 [3.3–13.2]	8.9 [5.4–14.3]	3.5 [1.4–9.2]	<0.0001
	(range)	(0.3–21.6)	(0.7–21.6)	(0.3–18.1)	
Cause of death	HSCT-related	137 (59.3)	65 (44.5)	72 (84.7)	<0.0001
*n* (%)	Disease-related	94 (40.7)	81 (55.5)	13 (15.3)	
	missing	2	1	1	
Total number of HSCT	Single HSCT	176 (75.5)	109 (74.1)	67 (77.9)	0.52
*n* (%)	Multiple HSCT	57 (24.5)	38 (25.9)	19 (22.1)	
Total number of HSCT (detailed)	1	176 (75.5)	109 (74.1)	67 (77.9)	Not tested
*n* (%)	2	45 (19.3)	32 (21.8)	13 (15.1)	
	3	9 (3.9)	6 (4.1)	3 (3.5)	
	4	3 (1.3)	0 (0)	3 (3.5)	

Abbreviations: HSCT = Hematopoietic Stem Cell Transplantation; Answers are described as number (percentage- by column, conditionally to the disease) or median [IQR].

**Table 3 children-08-00615-t003:** Survey respondents characteristics.

Questions	Responses	All Respondents (*n* = 98)	Physician (*n* = 42)	Nurse (*n* = 40)	Affiliated (*n* = 16)
Years working in current function *n* (%)	0 to 5	19 (21.6)	7 (20)	7 (18.9)	5 (31.2)
	5 to 10	14 (15.9)	7 (20)	4 (10.8)	3 (18.8)
	10 to 15	15 (17)	4 (11.4)	6 (16.2)	5 (31.2)
	15–20	14 (15.9)	5 (14.3)	7 (18.9)	2 (12.5)
	>20	26 (29.5)	12 (34.3)	13 (35.1)	1 (6.2)
	missing	10	7	3	0
Age (in years) *n* (%)	25–35	16 (16.8)	4 (9.5)	6 (16.2)	6 (37.5)
	35–40	12 (12.6)	7 (16.7)	4 (10.8)	1 (6.2)
	40–55	50 (52.6)	20 (47.6)	22 (59.5)	8 (50)
	55–60	11 (11.6)	6 (14.3)	4 (10.8)	1 (6.2)
	>60	6 (6.3)	5 (11.9)	1 (2.7)	0 (0)
	missing	3	0	3	0
Sex *n* (%)	Female	78 (79.6)	26 (61.9)	37 (92.5)	15 (93.8)
	Male	20 (20.4)	16 (38.1)	3 (7.5)	1 (6.2)

Answers are described as numbers (percentage by column, conditionally on the profession).

## Data Availability

The data are available on reasonable request from the authors.

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
