# Peer review of "Specialized Pediatric Palliative Care Services in Pediatric Hematopoietic Stem Cell Transplant Centers"

_children, 2021, doi:10.3390/children8080615_

Round 1
Reviewer 1 Report
Thank you for the opportunity to review your manuscript. This is impactful work that has the potential to add important knowledge to the fields of pediatric palliative care and pediatric HSCT. My enthusiasm for this paper is dampened by the lack of a coherent structure of the introduction, lack of clear language in the results section, consideration of only univariate associations, and the need for more contextualization of the results in the existing literature on this topic. Moreover, to target the focus of this paper, the authors might benefit from only focusing on the retrospective review in this analysis and focusing on the survey results in a separate manuscript. Specific comments are provided below for consideration to help strengthen the manuscript
Introduction:
- In paragraph 1, the language used does not make it clear that you are referencing other studies. Examples include the statement starting on line 48 (“Pediatric patients . . .) and the statement starting on line 52 (“The child’s condition . . .). This makes it unclear if you are talking about your own work or the work of others. Please include statements clarifying these results are from previous studies.
- In paragraph 2, authors should highlight/focus on importance of location of death as this is one of the major focuses of the article. There are several articles in the adult and pediatric literature showing that (1) palliative care involvement can improve rates of concordance between patient/family goals and location of death (Moody et al. 2020. J Pain Symptom Manage. PMID 32315752), and (2) achievement of goal-concordant end of life care, specifically location of death, is important (Johnston et al. 2021. Cancer. PMID 33784408; Ananth et al. 2021. J Pain Symptom Manage. PMID 33556497)
- In paragraph 3, there should be more of a transition from your first aim (exploring place and cause of death) and your second aim. As written, the transition from paragraph 2 to 3 is abrupt.
- In general, the introduction could benefit from a more structured approach to build to the authors primary aims of (1) describing place and cause of death of children who died following HSCT and (2) characterizing HCPs perceptions of the role of SPCCS in clinical HSCT practice.
Methods:
- Line 105: should read “pediatric HSCT physicians.”
- Line 117: should read “physicians, nurses, and other HCPs”
Results:
- Lines 138-145: For comparisons of location of death for those (a) dying from transplant complications or disease-related complications and (b) with malignancy or non-malignancy, these numbers should be reported as %’s of the total number of patients within each subgroup (transplant complications/disease complications, malignant/non-malignant) who died in each location with the cause of death/disease group as the denominator, not %’s of patient’s with given disease dying in each location with the location as the denominator.
- Lines 141-142: The statement that “most pediatric patients with a malignant disease died outside the hospital compared to patients with non-malignant disease” is not clear, as most patients with malignancy died inside the hospital (only 29 of 147 patients with malignant disease died at home. Could say something to this effect, “The majority of patients that died in the hospital died of HSCT-related causes. The majority of patients that died outside of the hospital had a malignant disease.”
- Should be Figure 1A if keeping figures 1B and 1C.
- The authors could consider removing figures 1B and 1C. The information is made clear in Table 1. Additionally, the way this data is presented in figures 1B/1C is somewhat misleading as the y axis (0-100%) is scaled differently for each place of death as variable numbers of patients died at each location. Visually, this is confusing. For example, Figure C shows the 91% of N=32 patients that died at home/hospice had a malignant disease, but this is only 29 patients. While 53% of N=88 patients that died in the PICU had a malignancy looks like a lesser value, this in fact corresponds to 47 patients.
- Figure 2: should arrange services in order from most to least important to make this figure clearer. Also advise simplifying the legend (examples: SDM = shared decision making could be combined into a single line, would spell out support in the figure).
Discussion:
- All associations presented in this analysis are univariate. Did the authors consider a multivariate model, perhaps with outcome as “in hospital vs out of hospital” and predictor variables as disease, cause of death, and time between last HSCT and death?
- Did the authors consider looking at single transplant vs multiple transplants as an independent variable that might be associated with location of death? Could this contribute to some of the differences observed between patients transplanted for malignant disease (perhaps more likely to have multiple transplants) and those transplanted for non-malignant disease?
- Lines 278-279: Advise including information from other studies/institutions about other known strategies for influencing rates/timing of pediatric palliative care consultation.
- As stated in summary comments, authors might consider separating the current manuscript into two papers: one focused on a strengthened analysis of the retrospective review, and another focused on the survey results.
Reviewer 2 Report
Thank you for this 2 part study which audited the death of children following HSCT and surveyed the opinions HCP on SPPC. The audit findings were expected and in-keeping with the clinical reality for children dying post HSCT. The opinions of HCP were very positive but at odds with timing of referrals but were, again, not unexpected.
The discussion of the results as it stands is a bit light. It could be improved and made more robust by referencing the wider literature of pediatric cancer not just that of HSCT/BMT. There are also a few relevant articles from 2019 and 2020 on the subject matter that have not been referenced.
I would also suggest the discussion on limitations include comment on how the survey was unable (probably due to survey design) to gain insight into the triggers for HCP referral or determine why there was such a discrepancy between perceived value of SPPC and referral rates. This discrepancy should have one or two paragraphs devoted to it in the discussion and supported by references.
Likewise this applies to the comments on early integration and advance care planning which may benefit from an exploration of standards in HSCT/BMT and pediatric cancer care.
Round 2
Reviewer 1 Report
We appreciate the authors efforts to improve this manuscript and the paper is greatly improved. There remain, however, many grammatical, sentence structure, and tense usage issues, which are distracting at times. Below are 4 examples with advice on how the sentences/grammar could be improved, though there are several others not listed below.
- Line 68: “Integration of palliative care during hospitalization for HSCT has proved beneficial in adult care . . .”
- Line 78-81, run on sentence. Could consider: “In contrast, several studies showed that transplanted children, adolescents, and young adults who received pediatric palliative care were more likely to die outside the PICU; this reflects an impact of PPC involvement on a patient’s location of death. Additionally, those who received PPC were also less likely to have intervention-focused care, and more likely to have opportunities for EOL communication and advance preparation.”
- Line 93-94: “ A Specialized pediatric palliative care services (SPPCS) could support an early palliative care approach.”
- Line 527-533: these sentences need a more complete restructuring. Could consider, “The literature describes several barriers to accessing palliative care in pediatric oncology at the level of policy, the health system, organizations, and individual providers. For individual providers, these barriers include a lack of knowledge, discomfort with speaking about death, cultural differences, and a lack of time or an established process to integrate palliative care services like ACP.”
Additionally, in the paragraph from lines 384-395, the use of different denominators when describing the percentages of various subgroups of patients, e.g. “most of the patients who died outside of the hospital had a malignant disease (n=29; 19.7%)”, is confusing. This is similar to the confusion caused by the previously included figures. There are many statements that state that “most” of the patients in a particular subgroup fell into another subgroup, but the provided proportion/percentages are well under 50%. I think it might be best to use this paragraph to summarize the significant differences seen in Table 1, but not include as many numbers/percentages as this makes the language confusing.
Author Response
We thank the reviewer for carefully reviewing our manuscript and we highly appreciate the suggestions to improve our work. We have revised our manuscript based on these suggestions. We rephrased the suggested sentences and critically revised the manuscript on grammar, sentence structure and tens use.
Please see the attachment.
